# CT-Based Radiomic Analysis May Predict Bacteriological Features of Infected Intraperitoneal Fluid Collections after Gastric Cancer Surgery

**DOI:** 10.3390/healthcare10071280

**Published:** 2022-07-10

**Authors:** Vlad Radu Puia, Roxana Adelina Lupean, Paul Andrei Ștefan, Alin Cornel Fetti, Dan Vălean, Florin Zaharie, Ioana Rusu, Lidia Ciobanu, Nadim Al-Hajjar

**Affiliations:** 13rd Department of General Surgery, “Iuliu Haţieganu” University of Medicine and Pharmacy, 400012 Cluj-Napoca, Romania; drpuiavr@gmail.com (V.R.P.); confetti_ro@yahoo.com (A.C.F.); valean.d92@gmail.com (D.V.); florinzaharie@yahoo.com (F.Z.); na_hajjar@yahoo.com (N.A.-H.); 2General Surgery Department, “Octavian Fodor” Gastroenterology and Hepatology Regional Institute (IRGH), 400162 Cluj-Napoca, Romania; 3Histology, Morphological Sciences Department, “Iuliu Haţieganu” University of Medicine and Pharmacy, 400006 Cluj-Napoca, Romania; 4Department of Biomedical Imaging and Image-Guided Therapy, General Hospital of Vienna (AKH), Medical University of Vienna, 1090 Wien, Austria; stefan_paul@ymail.com; 5Anatomy and Embryology, Morphological Sciences Department, “Iuliu Haţieganu” University of Medicine and Pharmacy, 400006 Cluj-Napoca, Romania; 6Department of Radiology, Emergency County Hospital, 400006 Cluj-Napoca, Romania; 7Department of Patology, “Octavian Fodor” Gastroenterology and Hepatology Regional Institute (IRGH), 400162 Cluj-Napoca, Romania; ioana.russu@yahoo.com; 8Gastroenterology Department, “Octavian Fodor” Gastroenterology and Hepatology Regional Institute (IRGH), 400162 Cluj-Napoca, Romania; ciobanulidia@yahoo.com

**Keywords:** infected peritoneal collections, computed tomography, bacteriology, texture-based analysis, gastric cancer, surgery

## Abstract

The ability of texture analysis (TA) features to discriminate between different types of infected fluid collections, as seen on computed tomography (CT) images, has never been investigated. The study comprised forty patients who had pathological post-operative fluid collections following gastric cancer surgery and underwent CT scans. Patients were separated into six groups based on advanced microbiological analysis of the fluid: mono bacterial (*n* = 16)/multiple-bacterial (*n* = 24)/fungal (*n* = 14)/non-fungal (*n* = 26) infection and drug susceptibility tests into: multiple drug-resistance bacteria (*n* = 23) and non-resistant bacteria (*n* = 17). Dedicated software was used to extract the collections’ TA parameters. The parameters obtained were used to compare fungal and non-fungal infections, mono-bacterial and multiple-bacterial infections, and multiresistant and non-resistant infections. Univariate and receiver operating characteristic analyses and the calculation of sensitivity (Se) and specificity (Sp) were used to identify the best-suited parameters for distinguishing between the selected groups. TA parameters were able to differentiate between fungal and non-fungal collections (ATeta3, *p* = 0.02; 55% Se, 100% Sp), mono and multiple-bacterial (CN2D6AngScMom, *p* = 0.03); 80% Se, 64.29% Sp) and between multiresistant and non-multiresistant collections (CN2D6Contrast, *p* = 0.04; 100% Se, 50% Sp). CT-based TA can statistically differentiate between different types of infected fluid collections. However, it is unclear which of the fluids’ micro or macroscopic features are reflected by the texture parameters. In addition, this cohort is used as a training cohort for the imaging algorithm, with further validation cohorts being required to confirm the changes detected by the algorithm.

## 1. Introduction

Intraperitoneal collections are common in the postoperative phase in abdominopelvic diseases and are easily depicted in almost all imaging modalities. All of these fluid collections, regardless of their content, have the potential to become infected and form an abscess. The identification of infected collections, as well as their underlying bacteriologic features, is critical because it drives treatment algorithms toward a more aggressive strategy, which may entail percutaneous or surgical draining in addition to an antibiotic prescription [1].

Despite recent breakthroughs in the treatment of infectious diseases, intraabdominal abscess continues to be a significant problem. Nevertheless, these infections can result in significant mortality and morbidity, particularly in specific patient subgroups. Although anaerobic gut microbes predominate, many intra-abdominal abscesses are polymicrobial. Antimicrobial therapy is essential for preventing and treating abscesses, but it cannot be used alone because it is frequently complicated by the abscess environment [2,3].

The most common imaging modality for detecting and assessing postoperative abdominal fluid accumulation is computed tomography (CT) [4]. Encapsulation, stranding of the surrounding fat, and the presence of gas within the collection have all been proposed to distinguish infected from noninfected collections [5]. These CT imaging characteristics are nonspecific because infected and noninfected fluid collections overlap extensively [6,7].

Fat stranding is an unspecific marker of inflammation that can occur in various circumstances, including ischemia, intestinal perforation, and, after surgery, in the absence of infection [8]. This was also confirmed in a recent study by Gnannt et al. [9], who discovered nearby fat stranding in 28% of sterile fluid collections. Fibrin, which acts as a defense mechanism for localizing and restricting peritoneal infections, is linked to the presence and enhancement of the wall surrounding fluid collections [10]. Fibrinolytic enzymes easily lyse fibrin in a healthy peritoneal cavity, but inflammation disables this mechanism. The presence of plasminogen activator inhibitor 1 in inflamed tissue may influence whether fibrin formed after a peritoneal injury is lysed or organized into fibrous adhesions [11,12]. These responses to infection can be variable and dependent on the host. Wall enhancement has conflicting findings. Allen et al. [13] found that it is not statistically significant for infection of loculated fluid collections, although other investigations have emphasized its value [14]. Among imaging measurements, wall enhancement has low specificity (50%), but high sensitivity (91%) (*p* < 0.001) and gas entrapment within the fluid collection has a low sensitivity (48%) but the highest specificity (93%) (*p* < 0.001). CT attenuation >10 HU showed intermediate sensitivity (74%) and specificity (70%) (*p* = 0.001) [15]. The sensitivity and specificity of these imaging findings are limited, making it difficult to distinguish sterile from non-sterile fluid collections. It is widely acknowledged that CT findings taken alone cannot predict the infection of a fluid sample with any certainty. This was previously demonstrated in a study of 92 patients with postoperative fluid collections, which found that the characteristics of gas entrapment and high-attenuation fluid (20 or larger HU) resulted in an average sensitivity of 83.4% and a poor specificity of 39.3% [13].

Although the imaging features suggestive of infection can be observed on computed tomography images, even in early stages, radiologists cannot make a further assumption about the biological/infective agent. However, these fluid collections’ biochemical, physical, and cytological features may characterize specific pathological processes [16]. These microscopic characteristics are expected to produce alterations in the density of CT images, but such changes are too subtle to be visible to human perception. In the last decade, quantitative imaging parameters have been introduced in radiology to increase diagnostic confidence and reduce the subjectivity of image interpretation [17,18,19]. One such computer-based application is radiomics, a quantitative approach to medical imaging that aims to improve the imaging-interpretation process using advanced mathematical analysis [20,21]. The underlying premise of radiomics is that medical images reflect “microscopic” disease-specific processes that are not visible to the human eye and are not accessible through traditional visual inspection [20,22].

This is a pilot study, conducted on a training cohort to highlight the changes between the radiomics variables. The aim of the study was to extract texture information from the post-operative intraperitoneal infected fluid collection. Our objective was to investigate whether the resulting parameters may help in the non-invasive distinction between different types/subtypes of fluid infections.

## 2. Materials and Methods

### 2.1. Patients

This is a retrospective study undertaken for the period January 2016 to December 2020 and involved 527 individuals who had gastric cancer surgery. Only 140 were referred to our radiology department for a CT scan of an intraperitoneal fluid collection after surgery. The hospital’s ethics committee (IRGH Cluj-Napoca, 14252/ 6 October 2021) approved this single-institution retrospective study, and a waiver of informed consent was obtained owing to its retrospective nature.

We searched the patients’ medical data in the hospital’s electronic system and extracted the following information: age and gender, underlying disease, type of gastric surgery, diabetes, the time interval between CT scan and attainment of infected fluid samples, intake of immunosuppressive drugs (glucocorticoids, cytostatics, drugs acting on immunophilins, interferons), body temperature, and blood sample, including C-reactive protein, as well as white blood cell counts.

Inclusion criteria for study were: patients diagnosed with gastric malignant tumours (with positive anatomopathological results), radical surgery (total or partial gastrectomy, multiorgan resections), presented infected intra-abdominal collections postoperatively (with positive bacteriology), fluid sample analysis being performed <20 days after CT–scan, and the absence of pathology that could have caused ascites.

Further inclusion criteria were as follows: microbiologic analysis of the collected fluid samples, body temperature >37.5 c, and blood sample including white blood cell counts (Leukocytes >15.000/µL) and C-reactive protein (CRP > 10 mg/L) within 24 h before drainage.

The exclusion criteria were: palliative surgery (*n* = 17), abdominal fluid collections in solid abdominal organs (*n* = 3), diameter under 30 millimetres of the collection (*n* = 28), missing laboratory parameter (*n* = 5), presence of imaging artifacts (*n* = 3), uninfected postoperative fluid collection (negative bacteriology; *n* = 44). Thus, we included a total of 40 patients after we used the given criteria.

All patients (*n* = 40) underwent analysis accordingly to the microbiological and DST examination and were structured as follows: group A: mono-bacterial/multiple-bacterial infections; group B: fungal/non-fungal infections; group C: multiple drug-resistance bacteria/non-resistant bacteria. We then divided the groups into subgroups in order to be compared one by one: mono-bacterial (*n* = 16) vs. multiple-bacterial (*n* = 24); fungal (*n* = 14) vs. non-fungal (*n* = 26); multiple drug-resistant bacteria (*n* = 23) vs. non-resistant bacteria (*n* = 17). Subgroups comprising the same individuals were not compared to each other, although subjects from the same main group were included in more subgroups (Figure 1).

### 2.2. Reference Standard

All patients suffered from malignant gastric tumours with positive anatomopathological results. The infected collections arose from patients who developed a postoperative fistula and were objectified by ultrasound/CT-guided puncture or surgical reintervention. The samples were collected as follows: 33 were obtained by ultrasound/CT-guided punctures and the other 7 were acquired during surgery. We initiated empirical antibiotic therapy at the time of reoperation. The choice of therapy took into account the severity of the case, previous antibiotic therapies, and local epidemiology. When a high risk of fungal infection was suspected, empirical antifungal medications were administered and adjusted to the results of peritoneal sample identification [23,24].

Microbiological examination of fluid samples is the gold standard for diagnosing infected fluid collections. Microscopic inspection and Gram staining are used to analyse samples. Agar plates are also employed as a culture medium for microorganisms. If microscopic and Gram stain results, as well as cultures, are negative, infection is deemed to be non-existent [25]. Multidrug-resistant (MDR) bacteria were defined as those resistant to three or more antimicrobial classes [26].

Immediately after ultrasound/CT-guided surgery, at least 5 mL of the drained fluid was then transported at room temperature to Microbiology Department for subsequent microbiological analyses (BACT/ALERT^®^ 3D). The same laboratory analysed all the fluid samples.

If leukocytes and bacteria were discovered on the Gram stain and/or the culture was positive, the fluid was declared infected by current standards (bacteria or fungi). The fluid collection was otherwise declared to be negative (i.e., noninfected).

The most frequent bacteria in the mono-bacterial group were staphylococcus aureus (*n* = 5). In the multiple-bacterial infection group the most frequent bacteria accounted for were: Gram-negative aerobic bacteria (*Escherichia coli*, *n* = 4; *Klebsiella* spp. *n* = 4), non-fermenting Gram-negative bacteria (*Pseudomonas aeruginosa*, *n* = 3), Gram-positive aerobic bacteria (*Enterococcus* spp. *n* = 6, *Streptococcus* spp. *n* = 2), and anaerobic bacteria (*Bacteroides* spp. *n* = 3). In fungal infections, the most frequent incriminated pathogen was *Candida albicans* (*n* = 9).

### 2.3. Image Acquisition

The Siemens Somatom Sensation with 16 slices was used for all CT scans (Siemens medical solutions, Forchheim, Germany). The CT scan spanned the area between the liver’s dome and the ischial tuberosity attachment. The CT scan was performed at 120 kV, 200 mAs, with a slice thickness of 3 mm.

### 2.4. Image Interpretation

One radiologist evaluated each examination on a dedicated workstation (General Electric, Advantage workstation, 4.7 version). A single slice was chosen as the best indicator of the fluid content during the non-enhanced part of each study. The selected slices were retrieved in DICOM format after all examinations were anonymized (Digital Imaging and Communications in Medicine).

### 2.5. Texture Analysis

The radiomics approach consisted of four steps: image segmentation using regions of interest, feature extraction, feature selection, and prediction.

#### 2.5.1. Image Pre-Processing and Segmentation

A second researcher used MaZda version 5 texture analysis software to import each image. The ascitic fluid was included in a two-dimensional (2D) region of interest for segmentation by the same researcher (ROI). For the defining and positioning of each ROI, a semi-automatic level-set technique was used. The researcher planted a seed in the approximate middle of the fluid collection, and the software used gradient coordinates to automatically define the collection. A demonstration of the ROI definition and placement can be found in Figure 2. Before the extraction of texture parameters, the imported image’s grey levels were normalized based on the mean and three standard deviations of grey-level intensities to reduce the contrast and brightness variations (which could affect the true image textures).

#### 2.5.2. Feature Extraction

The MaZda software’s built-in tools conducted the feature extraction automatically. The grey-level histogram, the wavelet transformation, the co-occurrence matrix, the run-length matrix, the absolute gradient, and the autoregressive model were used to generate approximately 300 texture parameters for each ROI. The parameters are detailed in the table below (Table 1).

#### 2.5.3. Feature Selection

The MaZda programme allows the selection of the most discriminative features through several reduction techniques. First, to select only the features with the highest discriminative ability, the probability of classification error and average correlation coefficients (POE+ACC) reduction technique was applied [27,28]. The POE+ACC algorithm introduces features with strong discriminatory potential and has the least association with previously selected features. A POE+ACC approach introduces 10 features with the lowest POE+ACC using measures of both probability of classification error (POE) and average correlation coefficients (ACC) between chosen features [29].

#### 2.5.4. Class Prediction

In order to further investigate which of the remaining parameters were best suited to discriminate between groups, the absolute values of the previously selected parameters were compared between the groups using the Mann–Whitney U test. The compared groups were: fungal (*n* = 13) and non-fungal (*n* = 27) infections, mono-bacterial (*n* = 16) and multiple-bacterial (*n* = 24) infections and multiresistant (*n* = 24) and non-resistant (*n* = 16) infections.

The statistically significant level was set at a *p*-value of below 0.05. All texture parameters that showed univariate analysis results below this threshold were excluded from further processing. The receiver operating characteristic (ROC) analysis was performed, with the calculation of the area under the curve (AUC) with 95% confidence intervals (CI) for the parameters that demonstrated statistically significant results in the univariate analysis (*p* < 0.05). Optimal cut-off values were chosen using a common optimization step that maximized the Youden index. Sensitivity (Se) and specificity (Sp) were computed from the same data without further adjustments. If multiple parameters showed statistically significant results in the univariate and ROC analysis, a multiple regression method using the “enter” input model was applied to investigate which input features are independent predictors for a certain group and the combined diagnostic value of these features. This step-by-step feature selection method was used in previous texture analysis studies [29,30,31], and the resulted parameters demonstrated adequate discriminative ability. Statistical analysis was performed using a commercially available dedicated software, MedCalc version 14.8.1 (MedCalc Software, Mariakerke, Belgium).

## 3. Results

Of the 527 patients recorded in our department during the study period, 40 were retrospectively included (22 women, 18 men, mean age 68 years, age range 34–87 years). Subjects were structured according to the final bacteriology and DST results into three groups and six subgroups as follows: group A: mono-bacterial (*n* = 16, 10 men and 6 women, mean age 66 years, age range 49–68 years)/multiple-bacterial (*n* = 24, 8 men, 6 women, mean age 71 years, age range 55–87 years) infection; group B: fungal (*n* = 14, 9 men, 5 women, mean age 76 years, age range 56–83 years)/non-fungal infections (*n* = 26, 10 men, 16 women, mean age 58 years, age range 48–66 years); group C: multiple-drug-resistant bacteria (*n* = 23, 13 men, 10 women, mean age 72 years, age range 55–76 years)/non-resistant bacteria (*n* = 17, 7 men, 10 women, mean age 65 years, age range 52–68 years).

The mean time between the CT examination and the fluid sampling was 9.7 days (range: 2–19 days). Prior to surgery, 11 patients had chemotherapy (27.5%); the most frequent comorbidity was diabetes in 17 patients (42.5%). Anaemia was found in 32 patients (80%), mean value 9.9, range 6.3–10.7 dl/L. Blood transfusion (mean value 1.7 UI, range 1–5 UI) was necessary for three patients before surgery and five patients during or after surgery. All patients received prophylactic antibiotic therapy, from which 28 (70%) patients required empirical antibiotic therapy (EAT) that was started at the time of reoperation. 

The infected collections were treated as follows: 7 patients required surgery per primam; 33 patients were initially treated conservatively by ultrasound-guided puncture and/or endoscopy (gastrointestinal stents, vacuum therapy with endo-sponge), 22 patients successfully, while 11 needed subsequent surgery. ICU (Intensive Care Unit) management was needed in 11 patients (27.5%) with a mean admission time of 8 days (range 3–21 days).

Two features (ATeta3 and ATeta4) showed statistically significant results (both *p*-values of 0.02) when comparing fungi vs non-fungi. Both features were independent predictors for fungal patients, as demonstrated by the multivariate analysis’ results (ATeta3, *p* = 0.0288 and ATeta4, *p* = 0.0326). When comparing the mono and multiple-bacterial parameters, one feature (CCN2D6AngScMom) showed statistically significant results in the univariate analysis (*p* = 0.03). The univariate analysis results are displayed in Table 2. The ROC analysis results are displayed in Table 3 and Figure 3.

## 4. Discussion

Our current results showed that two parameters (ATeta3 and ATeta4) were statistically significant when comparing fungi vs. non fungi. Both parameters were independently associated with fungal infections. Teta parameter area features of the autoregressive model describe the grey-level dependency of one pixel on other pixels in the neighbourhood [32]. The autoregressive-model-based parameters computed the spatial relationship among neighbourhood pixels [33]. In other words, this technique relies on the premise that each picture in an image is linearly dependent on its neighbours and, therefore, its values can be estimated using the grey levers of its neighbours in a defined neighbourhood. The image’s texture characteristics can be obtained by manipulating the pixel’s estimate against its real value [34]. Each parameter, taken separately, allows measuring the degree of randomness/regularity in the direction corresponding to the pixel associated with this parameter. In other words, when a Teta parameter displays low values (near 0 in absolute values), it is due to high randomness in the textures for that parameter’s direction. In contrast, a regular texture is a texture having the estimated parameters characterized by high values (near 1 in absolute values) [35]. The ATeta3 parameter showed higher values for the non-fungal group, while the ATeta4 showed higher values for the fungal group. At first glance, these results may seem contradictory. However, the two parameters (ATeta3 and ATeta4) are obtained through different computational methods (e.g., they are looking at the same picture from different perspectives).

The only texture feature that showed statistically significant results when comparing mono and poly was the CN2D6AngScMom feature. This parameter represents the uniformity of distribution of grey levels in the image [36]. We obtained higher values of this parameter for mono-bacteria than for polybacteria. 

Contrast is a measure of the local variations present in an image. If there is a high amount of variation, the contrast will be high [36]. We obtained higher values of this parameter for multiresistant than for non-multiresistant infections.

Changes in the bacterial flora of the GIT (gastrointestinal tract) determine the type and severity of post-anastomotic leak infection. The stomach and upper bowel flora have 104 organisms per gram or less, the lower ileum has up to 108 organisms per gram, and the colon has up to 1011 organisms per gram, with the majority of these being anaerobes [37]. The low amount of organisms in the stomach is thought to be related to the stomach’s low pH, having a negative effect on organisms ingested. In the lower intestine, the contents of the gut gradually become alkaline. This change, the effect of bile and the decrease in oxygen tension in the lower intestine allow for the selection of bile-resistant organisms and an increase in the number of strict anaerobes. Patients with decreased stomach acidity, such as those with gastric cancer or those with a shorter GIT or anastomosis, have a higher number of organisms in the upper GIT [38]. Differences in peritoneal cavity cultures following perforations can be attributed to variations in the number of bacteria in the GIT [39].

Inflammatory stimuli in the peritoneal cavity cause the exudation of protein-rich fluid into the peritoneal space, as well as the release of chemoattractant factors by resident peritoneal macrophages, which attract circulating neutrophils and monocytes [40]. The expression of tissue factor on the surface of peritoneal macrophages triggers local activation of the coagulation cascade, resulting in the deposition of fibrin around the inflammatory focus and producing the wall of an abscess—a process that serves to wall the infection from the rest of the peritoneal cavity.

Onderdonk et al. proposed the concept of ‘bacterial synergism’ more than 30 years ago, claiming that the coexistence of *E. coli*, *Enterococci*, and *Bacteroides fragilis* in experimental peritonitis was always fatal [41]. Anaerobes are known for their virulence, and species, such as *Bacteroides*, have been found to produce substances that directly impair polymorphonuclear leukocyte functions in humans [42].

The characteristics of bacterial colonization of the upper GI tract, numerous species of bacteria (multiple microbian infections), and the known process of bacterial synergism will all amplify the inflammatory response within the peritoneal cavity. This augmented response at the peritoneal cavity level could result in textural analysis changes, explaining the findings in multiple-bacterial, fungal, and multiresistant infections.

Because of its ability to quantify heterogeneities in radiological images, texture analysis was used to better describe fluid collections [43]. It is well accepted that debris is the primary source of greater fluid-collecting attenuation [9]. Debris and other accompanying findings of an abscess or infected fluid accumulation may induce unusual textural changes, which could explain the observations.

We previously demonstrated that the quantitative imaging parameters could increase the confidence in diagnosing malignancy-related ascites. Both commonly used quantitative measurements, such as Hounsfield units [44], apparent diffusion coefficients [45], but especially TA-derived parameters on CT [46] and MRI [47], were demonstrated to function as a non-invasive criterion that can aid the discrimination of benign and malignant intraperitoneal fluid collections.

The potential use of texture analysis to reflect histology and microstructure of malign ascites has been thoroughly proved for oncologic imaging [43,45,46,48].

The role of TA in the differentiation between infected and non-infected intraabdominal collections after oncologic surgery has been thoroughly analysed in one of our previous studies, with favourable results [49]. The studies regarding this topic are scarce, with only two current studies focusing on the textural analysis in infected collections, emphasizing the need for further investigations. Meyer et al. found that texture analysis of CT images is not superior to traditional imaging findings in distinguishing infected from non-infected fluid collections [50]. However, there is a bias regarding the specificity of the aforementioned study in terms of case selection. The analysed collections are located intra- and extraperitoneally and thoracicly and include parenchymatous organs, therefore, increasing the heterogeneity of the collection distribution. In addition, the study focuses on various aetiologies in the collections, including malignancy, trauma, and infectious and vascular pathology.

Both our studies focused on a specific aetiology regarding infected/non-infected IPEs to attain a minimum level of bias. Our department is focused on the oncology of the gastrointestinal tract. Furthermore, there is a distinction between bacterial colonization in the upper gastrointestinal tract and lower gastrointestinal tract infections in terms of bacterial colonization. Furthermore, oncological patients have a weaker immunological response to infections than healthy people. This research could pave the path for more research on IPE collections in a variety of gastrointestinal diseases.

To our knowledge, no study involving textural analysis was performed to extract texture information from the post-operative intraperitoneal infected fluid collection in order to distinguish between different types/subtypes of infections. An advantage of this study is that all the images were extracted from the same examination protocol, the same sequence, and the same machine, providing a higher degree of homogeneity of images.

Patients are frequently treated empirically with broad-spectrum antibiotics while waiting for results that reveal the bacterial class and species causing the infection, as well as drug susceptibilities. Traditional diagnostic approaches for deep-seated infections frequently rely on tissue biopsies to gather clinical samples, which can be costly, risky, and subject to sampling bias. Furthermore, these procedures and outcomes can take several days to complete and may not always offer accurate results. Because of the time and effort necessary for optimal antibiotic selection, indiscriminate broad-spectrum antibiotic use has become a problem [51].

Bacterial drug resistance is promoted by nosocomial infections and indiscriminate use of broad-spectrum antibiotics, resulting in significant morbidity and mortality, particularly in hospitalized and immunocompromised patients. To reduce morbidity and mortality rates caused by bacterial infections worldwide, early diagnosis of disease and tailored antibiotic treatments are critical. Reliable pathogen-specific bacterial imaging techniques have the potential to provide early diagnosis and guide antibiotic treatments [26].

Texture analysis, if confirmed in a more extensive series of patients, could be instrumental in current practice, guiding practitioners to specific microbiological infectious causes of intraperitoneal collections that are difficult to detect, especially when there are few peritoneal modifications or a small amount of fluid.

Our study had several limitations. First, it may contain selection bias due to its retrospective design. Furthermore, patients were assigned to groups solely on the basis of microbiological investigation of the fluid, not on biochemical (protein, albumin, pH value) and physical (density, viscosity, colour, surface tension) parameters, which are heterogeneous, even among collections with the same underlying condition. Another limitation of the study is the absence of a validation cohort, due to the small number of patients. Further studies, including validation cohorts, are required to adequately assess whether the changes found in this article are significant.

## 5. Conclusions

As a result, texture analysis may help distinguish specific microbiological properties in different types of intraperitoneal infected collections. It revealed significant textural differences between fungal and non-fungal, and mono and multiple-bacterial groups. We found that multiresistant infections had higher values of one parameter than non-multiresistant infections.

## Figures and Tables

**Figure 1 healthcare-10-01280-f001:**
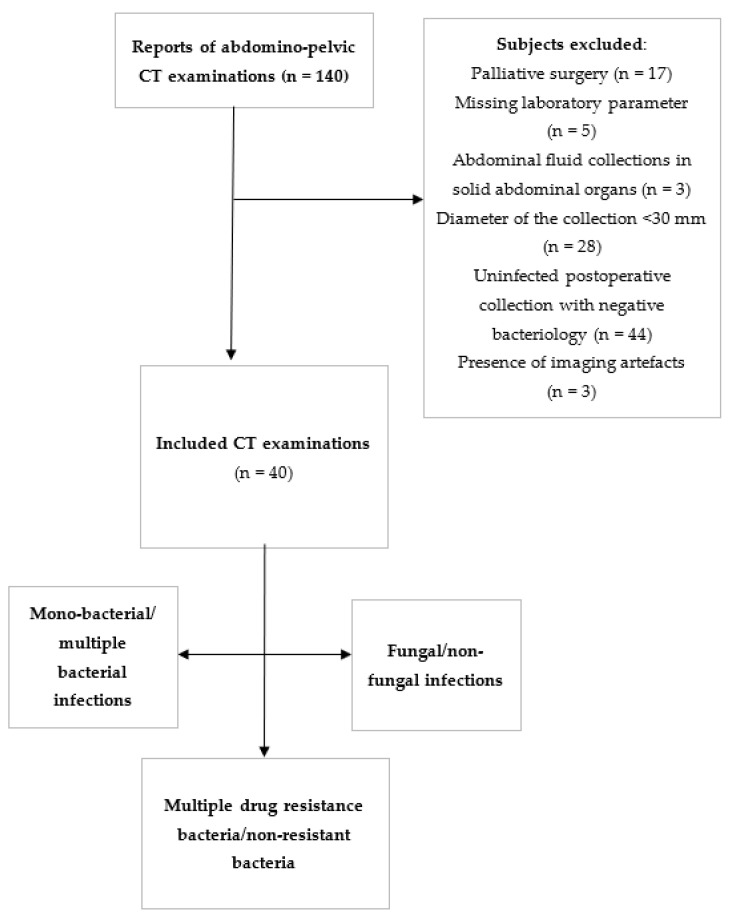
Flow chart. CT, computer tomography, patient groups.

**Figure 2 healthcare-10-01280-f002:**
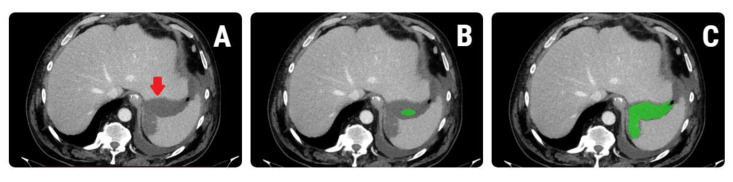
Synthetic demonstration of the region-of-interest (ROI) placement using (**A**) a computed tomography image of a 55-year-old patient with *Enterococcus* spp.—infected collection; the fluid collection is indicated with red arrow (**B**) the researcher placed a seed (green) within the collection and (**C**) the software automatically delineated the collection based on gradient and geometry coordinates.

**Figure 3 healthcare-10-01280-f003:**
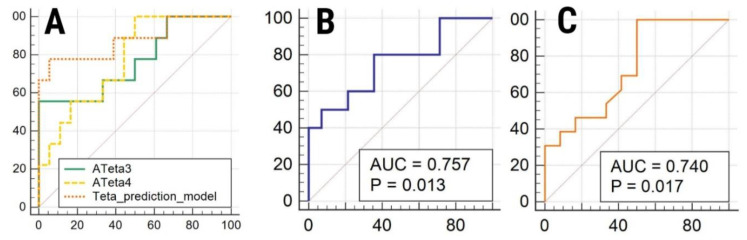
Display of ROC curves of (**A**) ATeta3, ATeta4 and Combined Teta model for the diagnosis of fungal infections; (**B**) CN2D6AngScMom for the diagnosis of fungal infections; (**C**) CN2D6Contrast for the diagnosis of multiresistant infections.

**Table 1 healthcare-10-01280-t001:** Texture parameters.

Parameters	Class	Computational Variations	Computation Method
Teta 1–4, Sigma	ARM	-	-
GrNonZeros, percentage of pixels with nonzero gradient, GrMean, GrVariance, GrSkewness, GrKurtosis	AR	-	4 bits/pixel
Perc.01–99%, Skewness, Kurtosis, Variance, Mean	Histogram	-	-
GLevNonU, LngREmph, RLNonUni, ShrtREmp, Fraction	RLM	4 directions	6 bits/pixel
InvDfMom, SumAverg, SumVarnc, SumEntrp, Entropy, DifVarnc, DifEntrp, AngScMom, Contrast, Correlat, SumOfSqs	COM	4 directions	6 bits/pixel; 5 between-pixel distances
WavEn	WT	4 frequency bands	5 scales

AR, Absolute gradient; RLM, Run Length Matrix; COM, Co-occurrence Matrix; ARM, Auto-regressive Model; WT, Wavelet transformation; Mean, histogram’s mean; Variance, histogram’s variance; Skewness, histogram’s skewness; Kurtosis, histogram’s kurtosis; Perc.01–99%, 1st to 99th percentile; GrMean, absolute gradient mean; GrVariance, absolute gradient variance; GrSkewness, absolute gradient skewness; GrKurtosis, absolute gradient kurtosis; GrNonZeros, percentage of pixels with nonzero gradient); RLNonUni, run-length nonuniformity; GLevNonU, grey level nonuniformity; LngREmph, long-run emphasis; ShrtREmp, short-run emphasis; Fraction, the fraction of image in runs; AngScMom, angular second moment; Contrast, contrast; Correlat, correlation; SumOfSqs, the sum of squares; InvDfMom, inverse difference moment; SumAverg, sum average; SumVarnc, sum variance; SumEntrp, sum entropy; Entropy, entropy; DifVarnc, the difference of variance; DifEntrp, the difference of entropy; Teta 1–4, parameters θ1–θ14; Sigma, parameter σ; WavEn, wavelet energy.

**Table 2 healthcare-10-01280-t002:** Univariate analysis results. Bold values are statistically significant.

**Fungal vs. Non-Fungal**	**Fungal**		**Non-Fungal**		** *p* ** **-Value**
	**Median**	**IQR**	**Median**	**IQR**	
ATeta3	0.23	0.18–0.43	0.42	0.36–0.52	**0.02**
ATeta4	0.05	0.01–0.16	−0.009	−011–0.04	**0.02**
CZ1D6Contrast	0.34	0.14–2.6	0.87	0.10–8.53	0.43
CN2D6Correlat	0.09	0.06–0.15	0.04	−0.002–0.12	0.10
RND6RLNonUni	140.61	59.45–653.45	498.60	38.69–1580.42	0.37
CH4D6Correlat	0.07	0.04–0.13	0.07	0.004–0.10	0.40
GD4Skewness	1.17	0.28–1.34	0.36	0.11–2.02	0.49
CV1D6Contrast	0.32	0.13–1.97	0.68	0.08–7.11	0.49
RVD6RLNonUni	95.78	43.84–571.62	506.96	22.78–1450.85	0.43
CH1D6AngScMom	0.35	0.13–0.73	0.19	0.01–0.80	0.40
**Mono vs. Multiple-bacterial**	**Monobacterial**		**Multiple-bacterial**		** *p* ** **-value**
	**Median**	**IQR**	**Median**	**IQR**	
CN5D6Correlat	0.08	0.02–0.15	0.01	−0.03–0.06	0.04
ATeta2	−0.18	−0.25–0.01	−0.15	−0.28–−0.03	0.75
CN2D6AngScMom	0.11	0.04–0.30	0.04	−0.003–0.08	**0.03**
WavEnHL_s-2	0.47	0.13–3.02	0.74	0.16–16.06	0.34
RVD6LngREmph	27.84	3.75–539.08	9.43	1.62–61.38	0.13
CH1D6Contrast	0.23	0.05–1.31	0.40	0.12–5.99	0.19
RZD6GLevNonU	223.30	117.44–505.92	169.84	90.70–228.98	0.25
RHD6LngREmph	35.00	4.31–513.91	11.68	1.69–89.29	0.15
ATeta4	−0.02	−0.12–0.04	0.03	0.01–0.14	0.12
Perc01	1001.5	113.5–1019.5	90.00	78.00–994.25	0.02
**Multi vs. Non-Multiresistant**	**Multiresistant**		**Non-Multiresistant**		** *p* ** **-value**
	**Median**	**IQR**	**Median**	**IQR**	
RND6GLevNonU	187.76	120.15–384.61	157.79	100.10–212.03	0.29
CH1D6DifVarnc	1.33	0.13–2.35	0.26	0.06–1.31	0.24
GD4Kurtosis	0.26	−1.08–0.50	0.19	−0.44–12.85	0.47
RHD6GLevNonU	165.87	91.77–369.11	131.18	89.00–184.78	0.24
ATeta1	0.51	0.33–0.58	0.59	0.36–0.69	0.34
CN5D6Correlat	0.01	−0.02–0.05	0.03	−0.01–0.09	0.43
WavEnLL_s-1	10,243.16	4182.69–12,237.74	10398	4286.96–16013.20	0.37
CN4D6Correlat	0.04	−0.04–0.06	0.03	0.00–0.09	0.53
Kurtosis	0.47	0.21–1.25	0.73	0.24–4.71	0.47
CN2D6*Contrast*	0.04	0.00–0.07	0.07	0.04–0.21	**0.04**

*p*-value, statistical significance value; bold values are statistically significant. IQR, interquartille range; Mean, histogram’s mean; Variance, histogram’s variance; Skewness, histogram’s skewness; Kurtosis, histogram’s kurtosis; Perc.01–99%, 1–99% percentile; GrMean, absolute gradient mean; GrVariance, absolute gradient variance; GrSkewness, absolute gradient skewness; GrKurtosis, absolute gradient kurtosis; GrNonZeros; RLNonUni, run-length nonuniformity; GLevNonU, grey level nonuniformity; LngREmph, long-run emphasis; ShrtREmp, (short-run emphasis; Fraction, the fraction of image in runs; AngScMom, angular second moment; Contrast, contrast; Correlat, correlation; SumOfSqs, the sum of squares; InvDfMom, inverse difference moment; SumAverg, sum average; SumVarnc, sum variance; SumEntrp, sum entropy; Entropy, entropy; DifVarnc, difference variance; DifEntrp, difference entropy; Teta 1–4, parameters θ1–θ4; Sigma, parameter σ; WavEn, wavelet energy.

**Table 3 healthcare-10-01280-t003:** Receiver operating characteristics’ analysis results.

Parameter	Sign.lvl.	AUC	J	Cut-Off	Se (%)	Sp
Fungi vs. non-fungi						
ATeta3	0.0137	0.765 (0.564–0.906)	0.5556	≤0.23	55.5 (21.2–86.3)	100 (81.5–100)
ATeta4	0.003	0.772 (0.571–0.91)	0.5	>−0.024	100 (66.4–100)	50 (26–74)
Combined Teta model	<0.0001	0.877 (0.717–1)	0.72	>0.49	77.78 (40.0–97.2)	94.44 (72.7–99.9)
Mono vs. poli microbian						
CN2D6AngScMom	0.0129	0.757 (0.541–0.907)	0.44	>0.05	80 (44.4–97.5)	64.29 (35.1–87.2)
Multirezistent vs. non multi						
CN2D6Contrast	0.0173	0.74 (0.528–0.893)	0.5	≤0.098	100 (75.3–100)	50 (21.1–78.9)

Sign.lvl., significance level; AUC, area under the curve; J, Youden index; Se, Sensitivity; Sp, Specificity; Between the brackets, values corresponding to the 95% confidence interval.

## Data Availability

Not applicable.

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
