# Peer review of "CT-Based Radiomic Analysis May Predict Bacteriological Features of Infected Intraperitoneal Fluid Collections after Gastric Cancer Surgery"

_healthcare, 2022, doi:10.3390/healthcare10071280_

Round 1
Reviewer 1 Report
The authors proposed to quickly distinguish bacterial and fungal infections after gastric cancer surgery of using CT-based radiomic analysis, which is an interesting and useful idea. To achieve this goal, the authors segmented the CT images, selected features, and made some statistical analysis. In general, the method and results are sound, and the paper is well-written. Here are some questions and I hope the authors could address them.
1. The authors mentioned a few patient selection criteria. I wonder if the (1) interval between surgery and CT scans and (2) the severity of infection were being taken into consideration? Do the authors think these can be a factor affecting the prediction results?
2. The authors mentioned that there are subtypes within both bacteria and fungi types. However, the authors simply divided all cases into 4 groups – fungal vs non-fugal and mono vs multiple bacterial. I wonder if differentiating between different subtypes would help improve the results?
3. In the method part (228-235), the authors mentioned a segmentation tool was used. Are there any parameters need to be specified in this tool? If yes, could the authors reveal the parameters used? How robust is the predict to different segmentation parameters?
4. Patient BMI usually affect image quality and therefore may affect textures. Is this factor being taken into consideration?
Reviewer 2 Report
In my opinion the manuscript is clear and well-structured. I have only few comments to add:
1) Line 53: please change "infectious infections" in infections or infectious diseases.
2) Line 119: since this is a retrospective study, I would not use the word "enrollment".
3)Line 440:please, specify better the limitations of the study, especially the second one mentioned.
Reviewer 3 Report
well written article. very interesting topic. I have only one question: why is there no comparison with nuclear medicine methods for infections? for example 18F-FDG PET or white blood cell (WBC) scintigraphy?Author Response
Please see the attachment.

Reviewer 4 Report
To authors and editors
CT-based radiomic analysis may predict bacteriological features of infected intraperitoneal fluid collections after gastric cancer surgery
The manuscript is very new interesting; nonetheless, there are some weak points that should be addressed.
1) The hospital’s ethics committee approved this single-institution retrospective study, and a waiver of informed consent was obtained owing to its retrospective nature. What's hospital? IRB number and date?
2) Figure 2. Please add arrow to indicate lesions?
3) Limitation is fine. What is further research suggestion to resolve these drawbacks?
4) Citations and references should be revised as MDPI format.
5) ROC figures are small. Please make it bigger.
6) Content is clear and effective without other concerns.
Sincerely
